# MSFDnet: A Multi-Scale Feature Dual-Layer Fusion Model for Sound Event Localization and Detection

**DOI:** 10.3390/s25206479

**Published:** 2025-10-20

**Authors:** Yi Chen, Zhenyu Huang, Liang Lei, Yu Yuan

**Affiliations:** 1School of Big Data and Information Industry, Chongqing City Management College, No. 151, Daxuecheng South Second Road, Shapingba District, Chongqing 401331, China; 2School of Big Data, Chongqing Vocational College of Economics and Trade, No. 1785, North Wuling Avenue, Qianjiang District, Chongqing 409000, China; 3School of Intelligent Technology and Engineering, Chongqing University of Science and Technology, No. 20, Daxuecheng East Road, Shapingba District, Chongqing 401331, China

**Keywords:** sound event localization and detection, attention mechanism, multi-task learning, feature fusion

## Abstract

The task of Sound Event Localization and Detection (SELD) aims to simultaneously address sound event recognition and spatial localization. However, existing SELD methods face limitations in long-duration dynamic audio scenarios, as they do not fully leverage the complementarity between multi-task features and lack depth in feature extraction, leading to restricted system performance. To address these issues, we propose a novel SELD model—MSDFnet. By introducing a Multi-Scale Feature Aggregation (MSFA) module and a Dual-Layer Feature Fusion strategy (DLFF), MSDFnet captures rich spatial features at multiple scales and establishes a stronger complementary relationship between SED and DOA features, thereby enhancing detection and localization accuracy. On the DCASE2020 Task 3 dataset, our model achieved scores of 0.319, 76%, 10.2°, 82.4%, and 0.198 in ${\mathrm{ER}}_{20}$,$F_{20}$, ${\mathrm{LE}}_{cd}$, ${\mathrm{LR}}_{cd}$, and ${\mathrm{SELD}}_{score}$ metrics, respectively. Experimental results demonstrate that MSDFnet performs excellently in complex audio scenarios. Additionally, ablation studies further confirm the effectiveness of the MSFA and DLFF modules in enhancing SELD task performance.

## 1. Introduction

SELD is a challenging research topic in the field of audio signal processing [1]. SELD is an integrated task that aims to simultaneously perform Sound Event Detection (SED) [2,3] and Sound Source Localization (SSL) [4,5]. The primary goal of SED is to identify specific sound events in audio signals, such as dog barking, car engines, or doorbells, while SSL focuses on determining the spatial position of these sound events, including their direction and distance. In this paper, we focus on estimating the Direction of Arrival (DOA) of sound events. With advancements in technology, SELD has found widespread applications in various fields, including audio surveillance, autonomous driving [6,7], construction activity monitoring [8], and smart homes [9]. By closely integrating sound event detection with localization, SELD not only enhances the performance of intelligent systems but also lays a solid foundation for future development, enabling diverse systems to achieve higher sensitivity and perceptual capabilities across various application scenarios.

With the continuous advancement of artificial intelligence, more sophisticated methods have been applied to SELD research. In 2010, Li Lu et al. proposed an SVM-based approach to evaluate the content of TV programs through audio analysis [10]. Later, in 2013, Saman Mousazadeh et al. achieved promising results in voice activity detection by combining Gaussian Mixture Models with the Laplacian operator. In 2016, Parascandolo et al. introduced a method for polyphonic sound event detection in real-life recordings using Bidirectional Long Short-Term Memory (BLSTM) recurrent neural networks (RNNs) [11]. Although these methods have achieved significant progress in their respective fields, they are primarily limited to detecting short-duration audio events. Challenges remain when applying them to long-duration and dynamic audio scenes.

As SELD research continues to advance, DCASE launched a challenge in 2020 focusing on long-duration audio event detection [12], aiming to address both SED and DOA over extended time periods. This task requires models to generate a temporal activation track for each target sound class based on multi-channel audio input and to output corresponding spatial trajectories when sound events are detected. DCASE subsequently introduced the SELDnet architecture [12], which extracts time–frequency features through convolutional layers and captures temporal dependencies using recurrent layers, enabling the collaborative processing of sound event detection and localization. However, SELDnet has certain limitations, prompting researchers to propose improved models. For example, Yin Cao et al. developed a Two-stage Network composed of Convolutional Neural Networks (CNNs) and Bidirectional GRUs [13]. The novelty of this network lies in first training the SED branch to detect event types and temporal information and then transferring the learned features to the DOAE branch for fine-tuning to enhance DOA estimation performance. In addition, Yin Cao and colleagues proposed the Event-Independent Network [14], which utilizes ResNet [15] to extract feature embeddings for both SED and DOA. These embeddings are fused and then fed into separate GRU networks for further processing. This fusion strategy effectively improves detection and localization accuracy and enhances the system’s performance in complex audio scenarios. However, these approaches still exhibit some limitations in spatial feature extraction, as they rely on only a few convolutional layers, which may lead to the loss of important spatial information. Moreover, they fail to fully leverage the complementarity between SED and DOA features, limiting the overall system performance. To address these issues, we proposed the MSDFnet model to overcome the shortcomings of existing methods. The main contributions of this work are as follows:MSFA: By introducing multi-scale convolutional layers, richer spatial features are captured at different scales, mitigating the issue of spatial feature loss caused by single-layer convolutions.DLFF: A dual-layer feature fusion network is designed to enhance the complementarity between SED and DOA features, thereby improving detection and localization accuracy.

In Section 2 we review previous approaches to SELD tasks and explore their limitations. In Section 3, a comprehensive overview of the MSFA module, the DLFF strategy, and the loss function is provided. In Section 4, the experimental setup is detailed, including the datasets, implementation details, and evaluation metrics. In Section 5, the results of comparison experiments with other models, along with ablation studies, are presented and analyzed. In Section 6, the proposed method is summarized, and potential directions for future research are discussed.

## 2. Related Work

In this section, we reviewed the evolution of SELD methods, which have transitioned from traditional approaches to deep learning, with continuous optimization of model structures to enhance our understanding of this work.

Early SED methods relied on feature engineering and statistical models, such as Gaussian Mixture Models [16], Hidden Markov Models [17], and SVM [18]. These methods extracted audio features manually (e.g., mel-spectrograms and MFCCs) and applied statistical models to classify sound events. While these approaches performed well in single-source, low-noise environments with short sound durations, they showed limitations in handling overlapping sound events in multi-source and noisy environments.

With the development of deep learning, artificial intelligence has made significant strides. CNNs [19,20,21] were introduced into SED tasks, marking substantial progress in SED technology. CNNs can automatically extract time–frequency features from audio signals, enhancing the automation of feature extraction. During this stage, research focused on designing network architectures suitable for audio data, such as using two-dimensional convolutional networks to process Log-Mel spectrograms and treating audio signals as images for recognition. This approach gradually became mainstream in DCASE challenges, achieving notable success in static sound source detection. Meanwhile, RNNs [22,23] were utilized to capture temporal dependencies in audio signals, combining with CNNs to form CRNNs [24], which achieved impressive results in DCASE and other challenges. CRNNs can capture both temporal and time–frequency features in audio, making them effective in handling dynamic audio scenarios and particularly strong in single-source detection and localization.

In recent years, to further enhance the adaptability of SELD tasks in multi-source scenarios, multi-label detection and multi-task learning strategies have gradually been introduced [25,26]. These methods enable the model to simultaneously detect and localize multiple sound sources in scenes with overlapping sound events. Although current SELD methods have achieved certain progress, limitations remain in feature extraction capacity and in fully leveraging the complementarity between tasks, resulting in suboptimal SELD performance.

Against this backdrop, our MSDFnet represents an effective attempt to address these challenges by efficiently extracting critical features for each task and effectively fusing features from different tasks, thereby making a positive impact on SELD performance.

## 3. Methods

In this study, we proposed the MSDFnet based on multi-task learning, designed for multi-source SED and DOA using multi-channel audio signals. As illustrated in Figure 1, the model architecture comprises multiple MSFA modules, a DLFF module, and a multi-task module. The input module first extracts Log-Mel spectrograms [27] and Intensity Vector [28] features from the audio signals, providing essential information for subsequent processing. The MSFA module aggregates features across different scales, enhancing the model’s adaptability to variations in time and frequency. The DLFF module further deepens and optimizes feature fusion through dynamic interaction among channel parameters. Additionally, the model incorporates a Multi-Head Self-Attention (MHSA) mechanism [29] at selected points to improve its ability to capture and process complex audio information effectively. Finally, the multi-task module outputs both classification and localization results, ensuring that the model excels in both multi-task and single-task environments.

### 3.1. Multi-Scale Feature Aggregation

We proposed the MSFA module, with its architecture illustrated in Figure 2. $X\in R^{B\times C\times H\times W}$ represents the input feature, where *B* denotes the batch size, *C* represents the number of channels, and *H* and *W* are the height and width, respectively. First, the module employs two layers of 2D convolution followed by batch normalization to extract spatial features. As shown in Equation (Equation 1),(1)$$X_{1}=\mathrm{ReLU}(\mathrm{BN}(\mathrm{Conv}2d\left(X,W_{1}\right)+b_{1})),$$
where $W_{1}$ is the convolution kernel, $b_{1}$ is the bias, BN represents batch normalization, and ReLU is the activation function. After applying the second convolutional layer, deeper-level features are extracted, as shown in Equation (Equation 2):(2)$$X_{2}=\mathrm{ReLU}\left(\mathrm{BN}\left(\mathrm{Conv}2d\left(X_{1},W_{2}\right)+b_{2}\right)\right),$$

In this study, each MSFA stage stacks two 3×3 convolutions (stride =1; padding =1) followed by average pooling to build a pyramidal multi-resolution hierarchy. On the SED branch, the three stages output 64/128/256 channels, respectively; on the DOA branch, the channel schedule is identical. The downsampling kernels are 2×2 for the first two stages and 1×2 for the deepest stage (time×frequency). Consequently, per stage we have two 3×3 conv layers whose kernel counts equal the stage output channels (e.g., Stage 1: 64+64; Stage 2: 128+128; Stage 3: 256+256 per branch).

Next, we introduced the EMA [30] module, an efficient multi-scale attention mechanism based on cross-spatial learning. EMA can extract audio features at multiple scales and adaptively fuse these features through weighted aggregation. This module captures the importance of features in different regions via the attention mechanism and dynamically adjusts the weights of each region [31], thereby enhancing the model’s feature representation capacity and robustness. The feature extraction operation of EMA can be denoted as $X_{3}=\mathrm{EMA}\left(X_{2}\right)$. Therefore, the expression of the MSFA module is presented in Equation (Equation 3).(3)O=AGP(EMA(ReLU(BN(Conv(X)))),
where Conv denotes a two-layer convolution operation, and AGP stands for the average pooling operation.

In this study, EMA aggregates features along three spatial ranges: global (1×1), row-wise (H×1), and column-wise (1×W) using adaptive average pooling. The pooled tensors are normalized (GroupNorm) and projected by one 1×1 and one 3×3 conv on the EMA branch. The EMA channel widths per stage are 8/16/32 (Stages 1–3), and attention weights are computed with a Softmax function along the last dimension and used to reweight and fuse the three spatial contexts. This design realizes cross-spatial learning without mixing multiple kernel sizes in the main branch.

### 3.2. Double-Layer Feature Fusion

To leverage the complementary nature of SED and DOA features, we propose a dual-layer feature fusion method for multi-task learning, as shown in Figure 3. Specifically, the module feeds the deep features of the SED and DOA tasks, extracted through the MSFA module, into the SE [32] module to dynamically compute the importance weights for each channel. The SE module operates through two steps, squeeze and excitation, effectively capturing global information and redistributing the weights across channels. This allows the model to focus on key information while suppressing irrelevant features, thereby enhancing its expressive power. The expressions are shown in Equations (4) and (5).(4)$$z_{c}=\frac{1}{H\times W}\sum_{i=1}^{H}\sum_{j=1}^{W}X_{c}\left(i,j\right),$$(5)$$s_{c}=\sigma \left(W_{2}\cdot \mathrm{ReLU}\left(W_{1}\cdot z_{c}\right)\right),$$

Equations (3) and (4) represent the squeeze and excitation operations of the SE module, respectively. Here, c=1,2,…,C denotes the channel index, *X* is the input feature, $X\in R^{C\times H\times W}$, $W_{1}$ and $W_{2}$ are the weight matrices of the fully connected layers, and σ represents the sigmoid activation function. After passing through the SE module, the SED and DOA features in this work can be represented as $s_{SED}^{SE}=\sigma \left(W_{2}^{\mathrm{SED}}\cdot \mathrm{ReLU}\left(W_{1}^{\mathrm{SED}}\cdot z^{\mathrm{SED}}\right)\right)$ and $s_{DOA}^{SE}=\sigma \left(W_{2}^{\mathrm{DOA}}\cdot \mathrm{ReLU}\left(W_{1}^{\mathrm{DOA}}\cdot z^{\mathrm{DOA}}\right)\right)$, respectively.

After processing through the SE module, the SED and DOA features undergo an initial fusion. The goal of this step is to aggregate the features from both tasks, enhancing the breadth of information. However, the fused features may dilute or obscure critical information, potentially impairing the model’s performance on both tasks. To address the potential loss of key features from the initial fusion, we designed a secondary fusion mechanism. Specifically, the output of the initial fusion is further combined with the features from the SE module through weighted fusion. This approach maintains the independence of features across tasks while ensuring the effectiveness of shared features, thereby improving the overall performance of the multi-task model, as shown in Equations (6) and (7):(6)$$X_{Fusion}=X_{SED}^{SE}+X_{DOA}^{SE},$$(7)$$X_{SED}^{Fusion}=X_{Fusion}+X_{SED}^{SE},X_{DOA}^{Fusion}=X_{Fusion}+X_{DOA}^{SE},$$
where Equation (Equation 6) represents the first feature fusion of SED and DOA, while Equation (Equation 7) denotes the secondary fusion for feature enhancement tailored to different tasks.

### 3.3. Multi-Tasking Module

In our multi-event sound localization and classification task, the traditional single-track [33] output cannot meet the requirement, as there may be overlaps of up to two sound events. Therefore, we introduced a multi-track output design to effectively handle the detection and localization of overlapping events. To further enhance the expressive power of the fused features, we incorporated the MSFA module into each track of every task, extracting more comprehensive features. The MSFA module aggregates feature information across different scales, providing the model with more representative feature expressions.

Moreover, SED and DOA tasks rely on the dynamic changes within the time sequence. To address this, we utilized the MHSA [29], which has demonstrated superior performance in temporal modeling across various fields, particularly in capturing long-range dependencies.

To reduce computational complexity, we employed average pooling along the frequency dimension after using the MSFA module for feature extraction. This operation effectively reduces the computational burden while retaining key features, ensuring that important information is not lost. This design achieves a balance between computational efficiency and model performance, ensuring that the model can be trained efficiently while accurately detecting and localizing multiple sound events. The expressions are shown in Equations (8) and (9).(8)$$X_{task}^{track}\left(b,c,t\right)=\frac{1}{F}\sum_{f=1}^{F}X_{task}^{MSFA}\left(b,c,t,f\right),$$(9)$$O_{task}^{track}=MultHead\left(X_{task}^{track}\right),$$
where track represents multiple sound events, task represents the sound detection task (SED or DOA), MSFA denotes the MSFA modules, MultHead represents the Multi-Head Attention, and *b*, *c*, *t*, and *f* represent the batch size, the number of channels, the time dimension, and the feature dimension, respectively.

### 3.4. Loss Function

In multi-event sound classification and localization, the SED task is formulated as a multi-label problem, as multiple sound events can occur simultaneously. For instance, a dog barking and a person speaking might overlap within the same time frame. In this study, we adopt cross-entropy loss [34] as the loss function for SED. The expressions are shown in Equation (Equation 10).(10)$${loss}_{SED}=-\frac{1}{N}\sum_{n=1}^{N}\sum_{t=1}^{T}y_{nt}\cdot \log\left(p({\hat{y}}_{nt})\right),$$
where $y_{nt}$ and ${\hat{y}}_{nt}$ are the true label and model predicted probability of the *n*-th sound event in the *t*-th frame, respectively.

The DOA task is usually regarded as a regression task, so we use the mean squared error loss [35] as the loss function for DOA. The expressions are shown in Equation (Equation 11).(11)$${loss}_{\mathrm{DOA}}=\sum_{n=1}^{N}\sum_{t=1}^{T}{∥{\hat{d}}_{nt}-d_{nt}∥}^{2},$$
where $d_{nt}$ and $\hat{d_{nt}}$ are the true value and model predicted value of DOA of the *d*-th sound event in the *c*-th frame, respectively.

The MSDFnet in this paper uses a joint loss function, which is a combination of SED loss and DOA loss. The weights are jointly optimized during training to obtain more accurate classification and localization performance. It is expressed as follows:(12)$${loss}_{\mathrm{SELD}}=\omega \cdot {loss}_{\mathrm{SED}}+\left(1-\omega \right)\cdot {loss}_{\mathrm{DOA}}$$
where ω is a weight parameter used to balance the losses of SED and DOA, which is set to 0.5 in this study.

## 4. Experiment

### 4.1. Dataset

This experiment uses the publicly available TAU-NIGENS Spatial Sound Events 2020 dataset (https://dcase.community/challenge2020/task-sound-event-localization-and-detection, accessed on 10 August 2025) provided by DCASE 2020 Task 3 [12]. The dataset has a total size of about 8.58 GB and is stored in WAV format. It consists of multi-channel audio synthesized to simulate realistic 3D soundscapes. Each audio clip is 1 min in length, recorded using 4 channels at a sampling rate of 24,000 Hz, corresponding to specific room setups. At any given time, there are at most two active sound sources, which may overlap or occur independently.

The dataset contains 14 sound event classes, namely alarm, crying baby, crash, barking dog, running engine, female scream, female speech, burning fire, footsteps, knocking on door, male scream, male speech, ringing phone, and piano. The number of annotated events for each class in the development and evaluation splits is summarized in Table 1.

In addition to the class labels, the dataset also provides detailed spatial information for each event, such as azimuth and elevation angles, which facilitates precise sound source localization.

1.The Log-Mel spectrogram [27] features were used to capture variations in the audio signal within the time–frequency domain, aiding the model in distinguishing different types of sound events.2.The Intensity Vector [28] features were employed to capture the spatial information of the audio signal, such as the direction and intensity of the sound sources, enhancing the model’s performance in spatial localization tasks.

The combination of these features enables the model to capture both temporal and spatial dimensions of the audio, supporting the goals of multi-task and multi-event sound source analysis.

### 4.2. Training Setup

In this experiment, we applied a 1024-point Hann window for the Short-Time Fourier Transform [36] on the audio data, with a 600-point frame shift to ensure smooth transitions between windows. The number of Mel bands was set to 256. Each audio clip was segmented into fixed 4 s chunks, ensuring no overlap between segments in the training and test sets.

The entire network was built using the PyTorch 2.0.1 with CUDA 11.8 on Ubuntu 22.04 framework. For optimization, we employed the AdamW optimizer. During the first 90 epochs, the learning rate was set to 0.0005, with a fixed random seed for reproducibility. In the following 10 epochs, the learning rate was reduced to 0.00005 to ensure more stable convergence during the later stages of training.

All experiments were conducted by training the model on the development set and testing on the evaluation set, with performance evaluated using 1 s segments. The training was carried out on a 12th Gen Intel^®^ Core™ i9-12900K CPU (Intel, Santa Clara, CA, USA) and an NVIDIA GeForce RTX 4090 GPU (NVIDIA, Santa Clara, CA, USA). To ensure the stability and reliability of the results, the final score was computed as the average of five independent trials.

### 4.3. Evaluation Metrics

In this study, we adopted the evaluation metrics from the DCASE 2020 SELD Challenge to comprehensively assess the model’s performance in the SELD task [12]. These evaluation metrics are divided into four categories, specifically evaluating the accuracy and robustness of SED and DOA estimation. Additionally, to gauge the model’s overall performance, we introduced SELDscore as a fifth integrated metric [1].

In the SED task, $ER_{20^{\circ }}$ and $F_{20^{\circ }}$ are two key metrics used to measure model performance. $ER_{20^{\circ }}$ evaluates the model’s false alarms and missed detections during sound event detection, while $F_{20^{\circ }}$ combines Precision and Recall to reflect the model’s accuracy in event detection. Both metrics are closely related to the precision of sound source localization: only when the localization error is less than $20^{\circ }$ will the model’s detection result be considered a correct match. The expressions are shown in Equations (13) and (14).(13)$$ER_{20^{\circ }}=\frac{\sum_{k=1}^{K}S\left(k\right)+D\left(k\right)+I\left(k\right)}{\sum_{k=1}^{K}N\left(k\right)},$$
where *S*, *D*, and *I* represent the incorrectly detected event labels, the events present but missed by the model, and the events incorrectly detected as present when they are not, respectively. *K* represents the total number of audio samples, and N(k) denotes the number of actual events in the *k*-th audio sample.(14)$$F_{20^{\circ }}=\frac{2\cdot \sum_{k=1}^{K}TP\left(k\right)}{2\cdot \sum_{k=1}^{K}TP\left(k\right)+\sum_{k=1}^{K}FP\left(k\right)+\sum_{k=1}^{K}FN\left(k\right)},$$
where TP, FP, and FN are the number of true positive examples, false positive examples, and false negative examples, respectively.

In the DOA task, $LE_{CD}$ and $LR_{CD}$ are important indicators for measuring the spatial positioning ability of the model. $LE_{CD}$ is the angular error between the predicted sound source position and the actual sound source position, which is used to quantify the accuracy of the model in spatial positioning. $LR_{CD}$ is used to evaluate the proportion of frames in which the predicted DOA of the model is consistent with the reference DOA in all time frames. This indicator calculates the proportion of time the model correctly detects and locates the sound source in each time frame. The expressions are shown in Equations (15) and (16).(15)$$LE_{CD}=2\cdot \arcsin\left(\sqrt{\frac{{\left(x_{G}-x_{E}\right)}^{2}+{\left(y_{G}-y_{E}\right)}^{2}+{\left(z_{G}-z_{E}\right)}^{2}}{2}}\right)\cdot \frac{180}{\pi },$$(16)$$LR_{CD}=\frac{TP}{TP+FN},$$
where $\left(x_{G},y_{G},z_{G}\right)$ and $\left(x_{E},y_{E},z_{E}\right)$ are the reference and predicted 3D coordinates, respectively.

In addition, the $SELD_{score}$ is calculated to aggregate all four indicators. Its expression is as follows:(17)$$SELD_{score}=\frac{SED_{score}+DOA_{score}}{2},$$
where $SED_{score}=\frac{ER_{20^{\circ }}+\left(1-F_{20^{\circ }}\right)}{2}$, $DOA_{score}=\frac{LE_{CD}/\pi +\left(1-LR_{CD}\right)}{2}$, and $SELD_{score}$ takes the above four metrics into account and reflects the overall performance of the model in the SELD task. In our experiments, we select the best model based on the $SELD_{score}$, and a lower score indicates a better overall performance of the model.

## 5. Results and Analysis

### 5.1. Baseline

**SELDnet** [12] This model serves as the baseline for DCASE 2020 Task 3, utilizing an improved version of SELDNet to perform both event detection and direction estimation. During training, an event-based masking technique is applied to enhance direction estimation by computing the loss only when events occur, thereby reducing unnecessary noise interference.

**Phan** [37] This approach uses a self-attentive CRNN architecture to propose a multi-task regression network. The model processes time–frequency domain data, extracts features through six convolutional layers, and encodes sequences with a bi-directional GRU. Finally, two branches output event activity and location trajectories.

**Cao_Surrey** [14] An event-independent end-to-end network is used for localization and detection of overlapping sound events. It utilizes frame-level alignment invariant training to address path alignment issues and improve detection accuracy for overlapping events. Additionally, Event Activity Detection is introduced as an auxiliary task to enhance interaction between SED and DOA.

**Nguyen_NTU** [38] In order to cope with complex audio signals with different sources and directions, it used a single-source histogram method for direction estimation and divided the direction estimation results into two independent parts, azimuth and elevation. This design makes the model more adaptable to multi-event detection and tracking scenarios.

**Park** [39] The method enhances the detection and localization capabilities of acoustic events by optimizing the loss function. To address the data imbalance issue, a range of loss functions were employed, including temporal masking loss, which enables the model to concentrate on the target event region for learning. Furthermore, period loss was introduced to augment azimuth prediction and enhance the precision of multi-output regression.

**DBAM** [40] The method uses a dual-branch attention network to process SED and DOA at the same time. DBAM uses the Conformer architecture to process local and global audio features in parallel. The convolutional branch extracts local details, and the self-attention branch captures long-range dependencies. The authors implement soft parameter sharing between the two sub-networks to facilitate the co-optimization of the SED and DOA tasks.

In order to verify the validity of the method, we use the same evaluation metrics as in Section 3.3 for comparison experiments. We select the official benchmark model of DCASE 2020 Task 3 and compare it with the excellent model of Phan, Cao, Nguyen, and Park’s team. In addition, a model that also uses the soft parameter sharing and attention module is selected for comparison.

### 5.2. Comparative Result

As evidenced by the results presented in Table 2, our proposed MSDFNet model demonstrates superior performance compared to the official SELDnet model and the models proposed by Phan, Park, and Cao_Surrey on the DCASE2020 Task 3 dataset across all five evaluation metrics. This superior performance can be attributed primarily to the more in-depth multi-scale feature extraction approach employed. This approach captures and preserves key features in the audio signal in a more comprehensive manner than traditional convolutional neural networks, thereby demonstrating excellent performance in the SELD task.

In the DOA task, our model performs similarly to the Nguyen_NTU and DBAM networks. The Nguyen_NTU network achieves the highest LRcd metric value of 82.7%, while our model achieves a comparable LRcd value of 82.6%. The DBAM network, on the other hand, achieves the lowest ${\mathrm{LE}}_{CD}$ metric value of 9.3°.

In terms of the overall combined performance of the SELD task, our method still achieves the best results. This advantage is mainly due to the introduction of a two-layer feature fusion method for different tasks, which effectively combines the complementary features of DOA and SED, and thus significantly improves the co-optimization of the model in the detection and localization tasks, demonstrating excellent overall performance.

We randomly selected two audio clips, mix080 and mix146, from the FOA evaluation dataset as examples. We converted the azimuth and elevation angles of the sound events into x, y, and z coordinates, which helps in directly quantifying and comparing the spatial errors between the two, making the analysis results more multidimensional and comprehensive. We plotted the sound event categories and their 3D coordinates in the audio clips, allowing for a direct validation of the SELD performance of the proposed method. Figure 4 show the reference and predicted values of categories and 3D coordinates in mix080 and mix146, respectively. The horizontal axis represents frames, corresponding to a time resolution of 100 ms, and the vertical axis represents categories or x, y, z coordinates.

As shown in Figure 4a, Mix080 is an audio clip without overlapping sounds. From the SED reference and predicted results, it is evident that our method accurately identifies the event categories in this audio. Additionally, the comparison between the X-, Y-, and Z-axis DOA reference and predicted values shows that our method performs excellently in horizontal localization, though some deviations are observed along the vertical axis.

In Figure 4b, Mix146 contains overlapping sounds. As indicated by the SED reference and predicted results, our method, despite some inaccuracies in multi-label classification in multi-source scenarios, still performs well for single-event detection. The comparison of the X-, Y-, and Z-axis DOA reference and predicted values similarly demonstrates superior horizontal localization; there are still some localization deviations along the vertical axis.

### 5.3. Ablation Experiments

We extend the ablation study beyond the module level to quantify the contribution of key sub-components. MSDFnet comprises two main modules, MSFA and DLFF, and we evaluate five variants: -DLFF (remove the entire DLFF), -MSFA (remove the entire MSFA), -MSFA(EMA) (remove only the EMA attention inside MSFA), -DLFF(Fusion) (remove only the secondary weighted fusion in DLFF), and MSDFnet (full).

Figure 5 reports results on ${\mathrm{ER}}_{20}$, $F_{20}$, ${\mathrm{LE}}_{\mathrm{CD}}$, ${\mathrm{LR}}_{\mathrm{CD}}$, and SELDscore. Removing any component degrades performance across all metrics. At the sub-component level, -MSFA(EMA) notably worsens ${\mathrm{ER}}_{20}$, $F_{20}$, and SELDscore, indicating that EMA is pivotal for MSFA’s multi-scale feature extraction; -DLFF(Fusion) lowers ${\mathrm{LR}}_{\mathrm{CD}}$ and raises SELDscore, showing that the secondary fusion is essential to strengthen SED–DOA complementarity after initial fusion. At the module level, -DLFF and -MSFA cause larger drops than their sub-component removals, confirming that MSFA provides strong feature extraction capacity while DLFF effectively exploits cross-task complementarities. Overall, the two modules and their internal designs (EMA and the secondary fusion) are complementary and jointly underpin the improvements on the SELD task.

### 5.4. Discussion

In this paper, we proposed MSDFnet, a novel multi-scale feature dual-layer fusion model for sound event localization and detection. By integrating the MSFA and the DLFF modules, MSDFnet effectively enhances the complementarity between SED and DOA estimation, leading to significant improvements in detection accuracy and localization robustness. Extensive experiments on the DCASE2020 Task 3 benchmark demonstrated that MSDFnet achieves state-of-the-art performance across multiple evaluation metrics. Ablation studies further verified the critical role of both the MSFA and DLFF modules in boosting overall system performance. These results confirm the effectiveness and potential of MSDFnet as a strong baseline for future SELD research.

## 6. Conclusions

In this paper, we introduce MSDFnet, a novel model for SELD tasks. Unlike prior approaches, our method utilizes a DLFF module to capture complementary SED and DOA information, boosting overall performance. Additionally, we design an MSFA module that captures audio features across scales, retaining essential information and enhancing the model’s capacity.

Extensive experiments on the DCASE2020 Task 3 benchmark validate MSDFnet’s superiority in SELD tasks. A detailed ablation study further confirms that each module significantly contributes to performance improvements. While MSDFnet shows strong results, some limitations remain. In overlapping sound scenarios, its performance can be suboptimal, likely due to limited training samples with overlapping sounds and feature similarities between classes. Additionally, z-axis localization in DOA tasks is moderately accurate, possibly due to limited vertical spatial information extraction.

For future work, we plan to increase model robustness with data augmentation and multi-source scenarios, explore 3D spatial feature extraction to improve z-axis localization, optimize the multi-task loss for SELD balance, and investigate cross-domain generalization to enhance real-world adaptability.

## Figures and Tables

**Figure 1 sensors-25-06479-f001:**
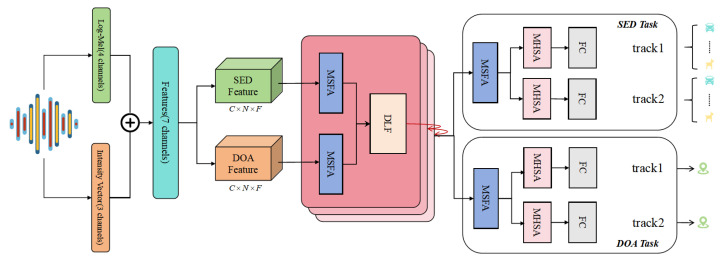
The structure of the proposed MSDFnet. ⊕ denotes the concatenation operation.

**Figure 2 sensors-25-06479-f002:**
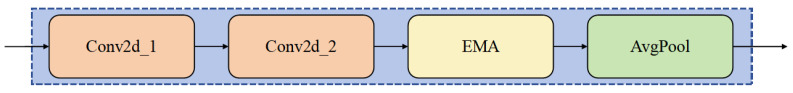
The structure of the proposed MSFA.

**Figure 3 sensors-25-06479-f003:**
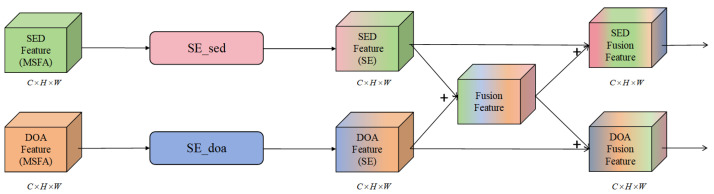
The structure of proposed DLFF. + denotes the element-wise addition operation.

**Figure 4 sensors-25-06479-f004:**
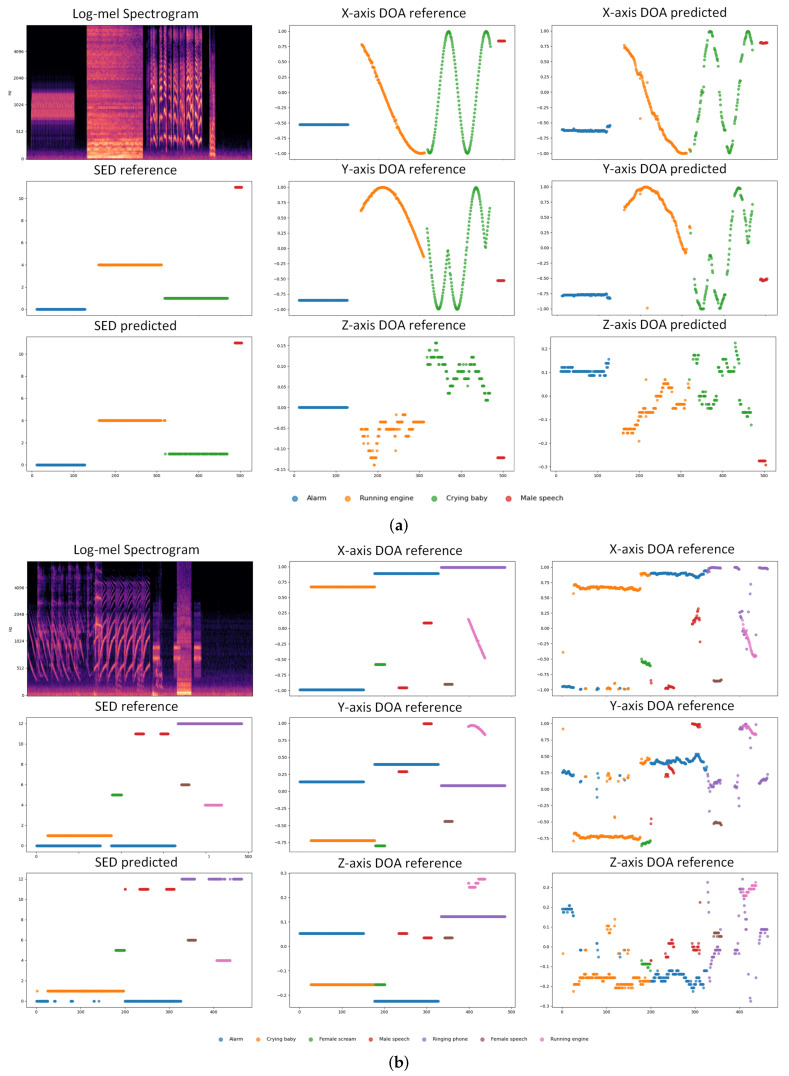
MSDFnet comparison of some of the data predictions in the evaluation set. In this, we converted the azimuth and elevation angles to x, y, and z coordinates. Different colors represent different categories, the horizontal axis represents frames, corresponding to a time resolution of 100 ms, and the vertical axis represents predicted or actual values. (**a**) The reference and prediction of x, y, and z coordinates and categories in audio Mix080 for the eval dataset. (**b**) The reference and prediction of x, y, and z coordinates and categories in audio Mix146 for the eval dataset.

**Figure 5 sensors-25-06479-f005:**
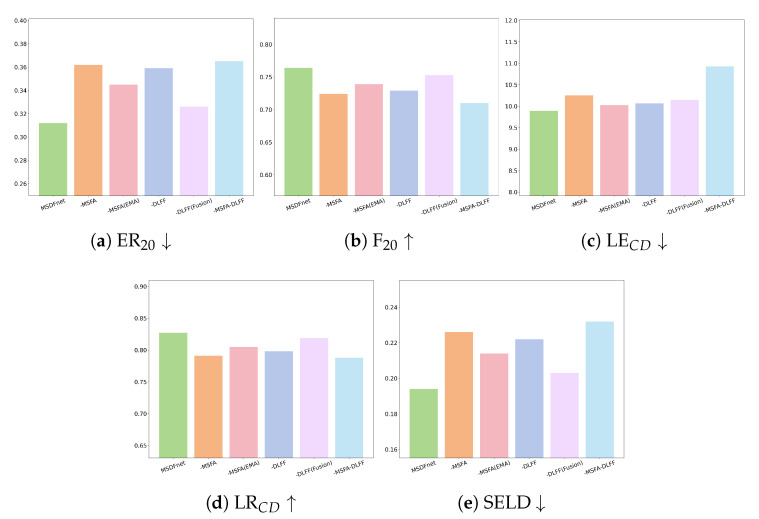
Ablation results across five metrics for MSDFnet variants. -DLFF, -MSFA, -DLFF-MSFA, -MSFA(EMA), and -DLFF(Fusion) denote removing the DLFF module, removing the MSFA module, removing both modules, removing only the EMA attention within MSFA, and removing only the secondary weighted fusion within DLFF, respectively; MSDFnet (full) is the complete model. Arrows indicate metric direction: ↓ means smaller is better, and ↑ means larger is better.

**Table 1 sensors-25-06479-t001:** Number of annotated sound events per class in the TAU-NIGENS Spatial Sound Events 2020 dataset.

Event Class	metadata_dev	metadata_eval	Total
Alarm	39,667	16,657	56,324
Crying baby	28,782	12,939	41,721
Crash	27,964	8083	36,047
Barking dog	22,301	6487	28,788
Running engine	42,986	12,225	55,211
Female scream	9841	2925	12,766
Female speech	5886	1908	7794
Burning fire	50,999	16,592	67,591
Footsteps	48,810	11,580	60,390
Knocking on door	5667	2613	8280
Male scream	9209	1174	10,383
Male speech	5591	2074	7665
Ringing phone	31,984	12,909	44,893
Piano	24,611	9054	33,665

**Table 2 sensors-25-06479-t002:** Comparison of different models based on various evaluation metrics.

Models	${\mathrm{ER}}_{20}↓$	$F_{20}↑$	${\mathrm{LE}}_{\mathrm{CD}}↓$	${\mathrm{LR}}_{\mathrm{CD}}↑$	SELD ↓
SELDnet	0.580	51.3%	18.3°	69.9%	0.367
Phan	0.490	61.7%	16.8°	81.9%	0.271
Park	0.430	65.2%	15.2°	72.4%	0.270
Cao_Surrey	0.363	71.2%	13.3°	81.1%	0.229
Nguyen_NTU	0.360	71.9%	12.1°	**82.7**%	0.220
DBAM	0.347	74.0%	**9.3°**	80.2%	0.214
MSDFnet (Our)	**0.312**	**76.4%**	9.89°	**82.7%**	**0.194**

↓ and ↑ indicate that smaller values are better and larger values are better, respectively. Bolded performance metrics indicate that the model performs best on these indicators.

## Data Availability

The TAU Spatial Sound Events 2020 dataset used in the experiments of this paper is available at https://dcase.community/challenge2020/task-sound-event-localization-and-detection (accessed on 10 August 2025).

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
