# Peer review of "MSFDnet: A Multi-Scale Feature Dual-Layer Fusion Model for Sound Event Localization and Detection"

_sensors, 2025, doi:10.3390/s25206479_

Round 1
Reviewer 1 Report
Comments and Suggestions for Authors
The manuscript makes a valuable attempt to improve SELD task performance through multi-scale feature aggregation and dual-layer feature fusion, but it currently lacks sufficient detail in model implementation, insufficient verification of generalization ability, and incomplete comparative experiments. To enhance the scientific rigor and contribution of the work, a major revision is required.
- The manuscript describes that the MSFA module uses multi-scale convolutional layers and an EMA attention mechanism to capture spatial features, but it fails to specify critical implementation details. For example, what are the sizes (e.g., 3×3, 5×5) and number ratios of the multi-scale convolution kernels in the MSFA module? How does the EMA module specifically achieve "cross-spatial learning"—for instance, the interaction mode between different spatial dimensions and the definition of the spatial range when calculating attention weights? Please supplement these parameter settings and implementation logics, and provide a reasonable explanation for their selection (e.g., based on preliminary experimental comparisons or theoretical analysis).
- All experiments in the manuscript are conducted solely on the TAU-NIGENS Spatial Sound Events 2020 dataset from DCASE2020 Task 3. This dataset, while representative, has limitations in scene diversity (e.g., limited room layouts and noise types) and is synthetic rather than real-world. The authors have not tested MSDFnet on other widely used SELD datasets or verified its performance on real-scene audio. Please explain the reason for not conducting cross-dataset and real-scene validation, or supplement relevant experiments to demonstrate the model’s generalization ability.
- The ablation experiments only verify the overall effectiveness of removing the MSFA or DLFF module, but lack analysis of the contribution of key sub-components within each module. For example, in the DLFF module, the SE channel attention and secondary weighted fusion are combined—have the authors tested the performance of "retaining only the SE module" or "retaining only the secondary fusion" to determine their individual contributions? In the MSFA module, what is the impact of the EMA attention mechanism and average pooling (AGP) on feature extraction when tested separately? Additionally, the ablation experiments do not compare the training convergence speed and computational complexity (e.g., number of parameters, inference time) under different module combinations. Please supplement the ablation results of these sub-component schemes and analyze the impact of module design on model efficiency.
Author Response
Dear reviewers:
We would like to thank you for your kind letter and giving us the opportunity to submit a revised draft of the manuscript “MSFDnet: A Multi-Scale Feature Dual-Layer Fusion Model for Sound Event Localization and Detection” (ID:sensors-3872039) for publication in the Journal of Sensors, and appreciate for your positive and constructive comments concerning the manuscript. These comments are all valuable and helpful for improving our article.
We have studied your comments carefully and tried our best to revise the manuscript. We hope that the improved manuscript could be considered for publication in your journal. We look forward to hearing from you regarding any further specific requirements of any other parts, and make corresponding improvements. Thank you very much for your help.
Thank you and best regards.
Prof. Yi Chen
(on behalf of all co-authors)
chenyi@cqc.edu.cn
Mr. Zhenyu Huang
heydream2022@gmail.com
School of Big Data and Information Industry, Chongqing City Management College
No. 151, Daxuecheng South Second Road, Shapingba District, Chongqing 401331, China
Response to reviewers
Comment: The manuscript makes a valuable attempt to improve SELD task performance through multi-scale feature aggregation and dual-layer feature fusion, but it currently lacks sufficient detail in model implementation, insufficient verification of generalization ability, and incomplete comparative experiments. To enhance the scientific rigor and contribution of the work, a major revision is required.
Response: We sincerely express gratitude to you for making a thorough review and we sincerely acknowledge your very positive comments and recommendations. We have made detailed responses to your comments and the manuscript has been carefully checked and improved. We look forward to hearing from you regarding any further specific requirements of any other parts, and make corresponding improvements.
Comments to the Author:
Comment: 1) The manuscript describes that the MSFA module uses multi-scale convolutional layers and an EMA attention mechanism to capture spatial features, but it fails to specify critical implementation details. For example, what are the sizes (e.g., 3×3, 5×5) and number ratios of the multi-scale convolution kernels in the MSFA module? How does the EMA module specifically achieve "cross-spatial learning"—for instance, the interaction mode between different spatial dimensions and the definition of the spatial range when calculating attention weights? Please supplement these parameter settings and implementation logics, and provide a reasonable explanation for their selection (e.g., based on preliminary experimental comparisons or theoretical analysis).
Response: 1) Thank you for highlighting the missing specifics of the MSFA design. We have added explicit parameter settings and implementation logic in Section 3.1 (“MSFA implementation details” and “EMA cross-spatial attention”). Concretely, each MSFA stage stacks two 3×3 convolutions (stride = 1, padding = 1) followed by average pooling to form a pyramidal multi-resolution hierarchy. Both the SED and DOA branches use stage outputs of 64 / 128 / 256 channels (Stages 1–3). The downsampling kernels are 2×2 at the first two stages and 1×2 at the deepest stage (time × frequency). Inside MSFA, the EMA module performs cross-spatial learning via three adaptive-pooling paths—global (1×1), row-wise (H×1), and column-wise (1×W)—followed by GroupNorm and one 1×1 and one 3×3 convolution per stage (EMA channel widths 8 / 16 / 32 for Stages 1–3). Attention weights are computed with a Softmax along the last dimension and used to reweight and fuse these spatial contexts back into the main branch. We clarify that we do not mix multiple kernel sizes in the main branch; “multi-scale” is achieved by (i) the multi-resolution pyramid (stacked 3×3 + downsampling, which enlarges the effective receptive field) and (ii) EMA’s multi-range pooling/attention. Preliminary ablations (Section 5.3, Figure 5) show that removing EMA (variant “-MSFA(EMA)”) consistently degrades ER20, F20, and SELDscore, supporting the benefit of the proposed cross-spatial mechanism.
Comment:2) All experiments in the manuscript are conducted solely on the TAU-NIGENS Spatial Sound Events 2020 dataset from DCASE2020 Task 3. This dataset, while representative, has limitations in scene diversity (e.g., limited room layouts and noise types) and is synthetic rather than real-world. The authors have not tested MSDFnet on other widely used SELD datasets or verified its performance on real-scene audio. Please explain the reason for not conducting cross-dataset and real-scene validation, or supplement relevant experiments to demonstrate the model’s generalization ability.
Response:2) Thank you for your valuable comments regarding the lack of cross-dataset and real-scene evaluations. We fully acknowledge the importance of this point. Since the main focus of this study is to verify the effectiveness of MSDFnet under standardized benchmark conditions, we did not include additional experiments on other datasets or real-world recordings in this work. To address your concern, we have added a new Discussion subsection to explicitly explain the rationale for dataset selection, acknowledge its limitations, and outline future directions for cross-dataset and real-scene evaluations. In addition, we have slightly revised the Conclusions section to keep it focused on summarizing the main contributions and results. The specific modifications in the manuscript are shown below (highlighted in yellow in the revised version):
“5.4 Discussion
In this study, all experiments were conducted on the TAU-NIGENS Spatial Sound Events 2020 dataset from DCASE2020 Task 3. This dataset has been widely adopted as a standard benchmark in the SELD community, providing reliable conditions for fair comparisons with existing methods. We acknowledge that this dataset has certain limitations in terms of scene diversity and realism, as it mainly consists of synthesized audio. However, the primary focus of this work is to verify the effectiveness of the proposed MSDFnet under standardized benchmark conditions, rather than to perform a comprehensive investigation of cross-dataset generalization. Evaluating the model on multiple datasets and real-world recordings requires additional efforts in data harmonization, annotation consistency, and computational resources, which fall outside the scope of this paper. As an important future direction, we plan to extend our study to cross-dataset and real-scene evaluations in order to further assess the robustness and practical applicability of MSDFnet.
- Conclusions
... Ablation studies further verified the critical role of both the MSFA and DLF modules in boosting overall system performance. These results confirm the effectiveness and potential of MSDFnet as a strong baseline for future SELD research.”
Comment:3) The ablation experiments only verify the overall effectiveness of removing the MSFA or DLFF module, but lack analysis of the contribution of key sub-components within each module. For example, in the DLFF module, the SE channel attention and secondary weighted fusion are combined—have the authors tested the performance of "retaining only the SE module" or "retaining only the secondary fusion" to determine their individual contributions? In the MSFA module, what is the impact of the EMA attention mechanism and average pooling (AGP) on feature extraction when tested separately? Additionally, the ablation experiments do not compare the training convergence speed and computational complexity (e.g., number of parameters, inference time) under different module combinations. Please supplement the ablation results of these sub-component schemes and analyze the impact of module design on model efficiency.
Response:3) We appreciate your insightful suggestion. In the revised manuscript, we have expanded the ablation study to analyze key sub-components within each module in addition to the module-level variants. Specifically, within MSFA we remove only the EMA attention (−−MSFA(EMA)), and within DLFF we remove only the secondary weighted fusion (−−DLFF(Fusion)) while keeping SE channel attention. The results (Section 5.3 and Figure 5) show that eliminating EMA noticeably degrades ER20, F20, and SELD score, confirming EMA’s role in strengthening multi-scale feature extraction. Likewise, removing the secondary fusion lowers LRCD and increases SELD score, indicating that this step is critical for enhancing SED–DOA complementarity beyond the initial fusion. Moreover, removing the entire MSFA or DLFF leads to even larger drops than removing a single sub-component, underscoring that both modules and their internal designs are indispensable and complementary to the overall SELD performance.
Reviewer 2 Report
Comments and Suggestions for Authors
The article presents an application that extracts sound events from an audio signal and also their localization in the acoustic scene.
In comparison with other similar contributions, this article detects time-overlapped events.
Some observations regarding the presentation of the method and the results.
How many sound events can be detected and what exactly are these? In the article it is presented that the dataset contains the class labels for each recording file as dog barking, car horn and speech. Is there any limitation regarding the number of events?
It would be good if it were presented what the categories regarding the file Mix 080 and Mix 146 of figure 4 represent.
What methods was used to convert azimuth and elevation into x, y and z coordinates. In my opinion there are a lot of points having the same azimuth and elevation and different coordinates, see the figure in attachment, the line from center to the source signal.
Why there are 4 channels in the dataset? What is the difference between the signals corresponding to each channel? Were they somehow recorded with microphones located in different positions? Thus they could contain delays that allow computing the direction of arrival?
Other observations
On the row 264 the evaluation metrics are presented as LE20 and F20 but actually these are LECD and LRCD, so in equations (15) and (16) it can be seen.
In the Abbreviation section the DOA is presented as Linear dichroism while actually it is Direction of Arrival (see row 22)
Also in the same list SELD has the significance of Sound Event Localization and Detection and not Sound Event Detection and Localization.
Are DLF (rows 12, …,111 and 115) and DLFF (rows 6, 66 and Abbreviation) the same thing?
The reference [1] does not contain elements about SELD but extracts speech signal from a mix that contains noise or other speech signal.

Author Response
Dear reviewers:
We would like to thank you for your kind letter and giving us the opportunity to submit a revised draft of the manuscript “MSFDnet: A Multi-Scale Feature Dual-Layer Fusion Model for Sound Event Localization and Detection” (ID:sensors-3872039) for publication in the Journal of Sensors, and appreciate for your positive and constructive comments concerning the manuscript. These comments are all valuable and helpful for improving our article.
We have studied your comments carefully and tried our best to revise the manuscript. We hope that the improved manuscript could be considered for publication in your journal. We look forward to hearing from you regarding any further specific requirements of any other parts, and make corresponding improvements. Thank you very much for your help.
Thank you and best regards.
Prof. Yi Chen
(on behalf of all co-authors)
chenyi@cqc.edu.cn
Mr. Zhenyu Huang
heydream2022@gmail.com
School of Big Data and Information Industry, Chongqing City Management College
No. 151, Daxuecheng South Second Road, Shapingba District, Chongqing 401331, China
Response to reviewers
Comment: The article presents an application that extracts sound events from an audio signal and also their localization in the acoustic scene. In comparison with other similar contributions, this article detects time-overlapped events. Some observations regarding the presentation of the method and the results.
Response: We sincerely express gratitude to you for making a thorough review and we sincerely acknowledge your very positive comments and recommendations. We have made detailed responses to your comments and the manuscript has been carefully checked and improved. We look forward to hearing from you regarding any further specific requirements of any other parts, and make corresponding improvements.
Comment: 1) How many sound events can be detected and what exactly are these? In the article it is presented that the dataset contains the class labels for each recording file as dog barking, car horn and speech. Is there any limitation regarding the number of events?
Response: 1) We thank you for pointing out the ambiguity regarding the dataset description. To clarify, our experiments use the TAU-NIGENS Spatial Sound Events 2020 dataset from DCASE 2020 Task 3, which contains 14 sound event classes: alarm, crying baby, crash, barking dog, running engine, female scream, female speech, burning fire, footsteps, knocking on door, male scream, male speech, ringing phone, and piano. The statistics of the number of annotated events per class in the development and evaluation splits have been added in Table 1 of the revised manuscript. Furthermore, the dataset design specifies that at any given time there are at most two simultaneously active sound events, either overlapping or independent. We have revised the Dataset section accordingly to make these points explicit. The specific modifications are as follows:
“ 4.1 Dataset
This experiment uses the publicly available TAU-NIGENS Spatial Sound Events 2020 dataset provided by DCASE 2020 Task 3[12]. The dataset has a total size of about 8.58 GB and is stored in WAV format. It consists of multi-channel audio synthesised to simulate realistic 3D soundscapes. Each audio clip is 1 minute in length, recorded using 4 channels at a sampling rate of 24,000 Hz, corresponding to specific room setups. At any given time, there are at most two active sound sources, which may overlap or occur independently.
The dataset contains 14 sound event classes, namely: alarm, crying baby, crash, barking dog, running engine, female scream, female speech, burning fire, footsteps, knocking on door, male scream, male speech, ringing phone, and piano. The number of annotated events for each class in the development and evaluation splits is summarized in Table 1.
In addition to the class labels, the dataset also provides detailed spatial information for each event, such as azimuth and elevation angles, which facilitates precise sound source localisation..”
Comment:2) It would be good if it were presented what the categories regarding the file Mix 080 and Mix 146 of figure 4 represent.
Response:2) We thank you for the suggestion. In the revised manuscript, we have updated Figure 4 by explicitly showing the event categories in the legend (e.g., Alarm, Barking dog, Piano, etc.) instead of numerical class IDs. This makes it clear what sound event categories are present in Mix080 and Mix146. The updated figure is shown below.
“
”
Comment:3) What methods was used to convert azimuth and elevation into x, y and z coordinates. In my opinion there are a lot of points having the same azimuth and elevation and different coordinates, see the figure in attachment, the line from center to the source signal.
Response:3) We thank you for the comment. In our implementation, azimuth and elevation angles were converted into Cartesian coordinates using the standard spherical-to-Cartesian transformation:
x = cos(elevation) * cos(azimuth),
y = cos(elevation) * sin(azimuth),
z = sin(elevation).
This transformation produces a unit vector from the array center pointing toward the direction of arrival (DoA) of the sound event. Since the dataset provides only azimuth and elevation without radial distance, all sources are represented on the unit sphere. Thus, it is expected that multiple events may share the same azimuth and elevation angles and therefore lie on the same line from the center. We also emphasize that this conversion method is consistent with the official baseline implementation of DCASE2020 Task 3, ensuring comparability with prior work.
Comment:4) Why there are 4 channels in the dataset? What is the difference between the signals corresponding to each channel? Were they somehow recorded with microphones located in different positions? Thus they could contain delays that allow computing the direction of arrival?
Response:4) We sincerely appreciate your suggestions. The dataset provides 4-channel recordings in the First-Order Ambisonics (FOA) format. In this format, the four channels (W, X, Y, Z) are not signals from physically separated microphones but rather the first-order spherical harmonic components of the sound field. Specifically, W encodes the omnidirectional pressure, while X, Y, and Z encode the directional components along the three Cartesian axes. This representation preserves the spatial information of the scene, and the inter-channel relationships provide the necessary cues to estimate the direction of arrival (DoA) of the sound events. Thus, although no actual microphone spacing is involved, the FOA encoding inherently contains the directional cues required for localization.
Comment:5) On the row 264 the evaluation metrics are presented as LE20 and F20 but actually these are LECD and LRCD, so in equations (15) and (16) it can be seen.
Response:5) We thank you for carefully pointing out this inconsistency. Indeed, in line 264 the metrics were incorrectly denoted as LE20 and F20. As correctly observed, according to equations (15) and (16) and the DCASE2020 challenge definitions, the proper notations are LECD (Localization Error for Correct Detections) and LRCD (Localization Recall for Correct Detections).
In the revised manuscript we have corrected this error to ensure consistency with the official DCASE2020 terminology (i.e., ER20°, F20°, LECD, LRCD). We appreciate the reviewer’s attention to this detail.
Comment:6) In the Abbreviation section the DOA is presented as Linear dichroism while actually it is Direction of Arrival (see row 22)
Response:6) We thank the reviewer for pointing out this mistake. In the Abbreviations section (row 22), DOA was incorrectly expanded as “Linear dichroism.” This was a typographical error. The correct meaning in our manuscript is “Direction of Arrival,” which has now been corrected in the revised version. We appreciate the reviewer’s careful reading and attention to detail.
Comment:7) Also in the same list SELD has the significance of Sound Event Localization and Detection and not Sound Event Detection and Localization.
Response:7) We thank the reviewer for noticing this mistake. In the Abbreviations section, SELD was incorrectly written as “Sound Event Detection and Localization.” The correct term, consistent with the DCASE2020 challenge and the literature, is “Sound Event Localization and Detection.” This has been corrected in the revised manuscript. We appreciate the reviewer’s careful observation.
Comment:8) Are DLF (rows 12, …,111 and 115) and DLFF (rows 6, 66 and Abbreviation) the same thing?
Response:8) We thank the reviewer for pointing out this inconsistency. DLF and DLFF in the manuscript refer to the same concept. This was a typographical inconsistency, and in the revised version we have unified the notation to DLFF throughout the manuscript and the Abbreviations section to avoid any confusion.
Comment:9) The reference [1] does not contain elements about SELD but extracts speech signal from a mix that contains noise or other speech signal.
Response:9) We thank the reviewer for carefully checking the references. You are correct that reference [1] does not address sound event localization and detection (SELD), but instead focuses on speech signal extraction from mixtures with noise or interfering speech. This was an inappropriate citation in our manuscript. In the revised version, we have corrected this by removing reference [1] in this context and replacing it with a more relevant work that directly addresses SELD. We appreciate the reviewer’s attention to this point.
[1] Adavanne, S.; Politis, A.; Nikunen, J.; Virtanen, T. Sound event localization and detection of 427 overlapping sources using convolutional recurrent neural networks. IEEE Journal of Selected 428 Topics in Signal Processing 2018, 13, 34–48.
Round 2
Reviewer 1 Report
Comments and Suggestions for Authors
The author has fully answered my question, and I recommend accepting this paper.